# Exploring the reasons for novice nurse educators' transition from practice to academia in Ghana

Timothy Tienbia Laari[1,2]*, Felix Apiribu[2], Philemon Adoliwine Amooba[2], Adwoa Bemah Boamah Mensah[2], Timothy Gazari[3], Joseph Kuufaakang Kuunibe[4], Gideon Awenabisa Atanuriba[5], Moses Haruna Akor[6]

1 Presbyterian Primary Health Care (PPHC), Bolgatanga, Ghana, 2 Department of Nursing, Faculty of Allied Health Sciences, College of Health Sciences, Kwame Nkrumah University of Science and Technology, Kumasi, Ghana, 3 Department of Mental Health Nursing, University for Development Studies, School of Nursing and Midwifery, Tamale, Ghana, 4 Midwifery Training College, Tumu, Ghana, 5 Tamale Regional/ Central Hospital, Tamale, Ghana, 6 Nursing Training College, Damongo, Ghana

* timlaari@yahoo.com

**Data Availability Statement:** All relevant data are within the paper and its Supporting Information files.

## Abstract

### Background

There is an increasing transition rate of experienced clinical nurses from practice to academia. When nurses transition from practice to academia for the right reasons, it culminates in job satisfaction and retention. Thus, understanding what attracts clinical nurses to academia is an important consideration for employing and retaining competent nurse educators. Yet, there are gaps in research about what motivates nurses to transition from practice to academia within the Ghanaian context. This study aimed to explore the reasons for novice nurse educators' transition from practice to academia in three Health Training Institutions in the Upper East Region of Ghana.

### Methods

This qualitative descriptive phenomenology study used a purposive sampling method to select 12 novice nurse educators. Data were collected using a semi-structured interview guide through individual face-to-face in-depth interviews. Written informed consent was obtained and interviews were audio-taped and transcribed verbatim. Data analysis was done manually guided by Colaizzi's method of data analysis.

### Results

Novice nurse educators transitioned from practice to academia because they were dissatisfied with their clinical nursing practice, they wanted more flexible work, they wanted to work autonomously, and they previously taught their clients in the clinical setting. Four themes emerged namely: (1) dissatisfied with clinical nursing, (2) quest for flexible work role, (3) quest for work autonomy, and (4) previous clinical teaching.

**Funding:** The author(s) received no specific funding for this work.

**Competing interests:** The authors have declared that no competing interests exist.

**Abbreviations:** NNEs, Novice Nurse Educators; HTIs, Health Training Institutions; MOH, Ministry of Health.

## Conclusion

The reasons for transitioning from practice to academia were mostly born out of novice nurse educators' previous negative experiences in the clinical setting which ought to be considered in the recruitment and retention of teaching staff to train the future nurses. There is the need to revise and implement a tutor recruitment policy that takes into account, what attracts clinical nurses to the academic setting.

## Introduction

Many Low and Middle-Income Countries (LMICs) face a serious shortage of nurses and midwives, a shortage that presents a critical constraint to the achievement of health and development goals, and scaling up educational programmes for nurses and midwives is one way to address this threat [1]. Scaling up nursing and midwifery education can transform population health outcomes but to do so, nations must increase not only the quantity but also the quality, and the relevance of nurse educators is critical to this agenda [1]. Quality nursing and midwifery education is a key driver for improved quality of healthcare services. There is a critical need for competent nurse educators to develop and implement evidence-based curricula geared at training the future nursing workforce [2]. Therefore, the World Health Organisation (WHO) recommends that nursing or midwifery training colleges should employ nursing or midwifery faculty members with relevant expertise in the clinical area and the ability to develop and teach nursing programmes [3].

Between 2001 and 2006, Ghana's Ministry of Health (MOH) expanded all existing Health Training Institutions (HTIs), set up 21 new HTIs, and the private sector also established 7 new nursing colleges. Programmes have been introduced for direct entry into midwifery, Health Assistant Clinical (HAC), and diploma in Community Health Nursing (CHN) programmes [4]. This policy of the MOH resulted in a 50% increase in admissions into HTIs since 2001. However, notwithstanding these gains in recent years, the capacities of the HTIs to train sufficient numbers to meet national requirement remains insufficient in terms of teaching staff [4]. The strategy to increase student intake without the corresponding recruitment of teaching staff has led to academic staff vacancies in the HTIs [4]. To seal these ensuing faculty vacancies, the HTI secretariat periodically recruits expert nurses at the hospital setting into various HTIs across the country as health tutors.

Most nurses in Ghana are trained in NTIs commonly known as Nursing Training Colleges (NTCs), some of which are affiliated with teaching hospitals [5]. At the time of data collection, there were a total of 118 HTIs in Ghana comprising 78 state-owned and 40 private-owned. These HTIs usually offer certificate, diploma, and post-basic diploma in diverse nursing programmes [6]. The nurse educators in these HTIs are tutors with first degrees who previously worked as professional nurses in the clinical setting. However, baccalaureate nursing programmes are offered in universities that award a bachelor degree upon completion, although several of those are private institutions [5]. The teaching staff in these universities are usually lecturers who have doctoral degrees. All successful graduates from certificate, diploma, or baccalaureate programmes qualify to write the nursing licensing examination administered by the Nursing and Midwifery Council of Ghana [5]. All certificate trained nurses are called auxiliary nurses while diploma and baccalaureate trained are professional nurses. Qualified professional nurses who work in public and private hospitals may later apply to the HTIs secretariat for recruitment as nurse educators and if selected, will make a transition into academia.

Making a transition from clinical nursing practice to academic nursing implies making a counterintuitive coming back to a beginner from proficient because having high proficiency in clinical nursing practice does not guarantee proficiency in the nursing academic role; therefore, taking the decision to transition from bedside to classroom requires cautious consideration of many private and professional factors [7]. The choice to pursue a role in academic nursing must prompt the clinical nurse to take a personal account and evaluate one's personal desired place of work and set of skills [8]. Therefore, to ensure nurses in the clinical setting are making the transition from the clinician to educator role for the correct reasons, expert nurse clinicians should reflect on what is motivating their decision to transition [9].

When nurses transition from the clinical setting to academia for the right reasons, it leads to job satisfaction and retention. A study in the United States of America (USA) found that clinical nurse experts transitioned from practice to academia because they had the desire to educate the next generation through teaching and believed that they had expert knowledge to share [10]. In a similar study in Massachusetts and New Hampshire, USA, the driving and encouraging forces that motivated the transition of nurses into the full-time faculty role included the strong love of teaching, financial incentives available at the faculty level, and the want of more flexible work schedule [11]. Related studies in the USA have also revealed analogous findings including the desire to teach students in an academic setting, work flexibility, and autonomy [12], ready to share expert knowledge with the next generation of nurses [13], wanting to impart the nurse-midwifery philosophy on students [14], and making a difference in the profession by influencing the next generation of nurses, more flexible work schedule, and career progression [15].

In Canada, the desire to teach, the excitement of seeing a student learn and succeed, contributing to the nursing profession by educating future nurses, sharing nursing knowledge with students, more flexible work schedule and vacation time, and seeing students learn and understand professional nursing were the reasons why Canadian nurses transitioned from practice to academia [16]. These studies demonstrate that nurses in the clinical setting often have varied reasons for transitioning from clinical nursing to academic nursing and a successful transition to the academic role largely begins with having a good reason. The transition begins when the nurse makes the decision to become a nurse educator and having good reasons for the transition yields to job satisfaction and intention to stay [15].

In Low and Middle-Income Countries (LMICs), few studies have explored the transitional experiences of NNEs, especially in the African continent. Notably, a study in South Africa on the experiences and mentoring needs of NNEs revealed that the lack of mentoring causes a difficult transition for NNEs from the nursing role into the nurse educator role [17]. Also, an integrative review on nursing education challenges and solutions in Sub-Sahara Africa revealed that NNEs had non-monetary expectations in nursing academia and that academia offered them an opportunity to develop themselves as nurse educators [18]. However, the reasons for NNEs' transition from practice to academia is still understudied in these LMICs.

In Ghana, the health training institution secretariat is mandated to identify and recruit proficient clinical nurses into HTIs across the country. As a result, more and more experienced nurses in the clinical setting are often moved into the academic setting without considering what motivate these expert clinicians to venture into nursing academia. While we know its importance, there is no existing literature on what motivates clinical nurses to transition to the academic role in Ghana. This paper reports the findings of one objective of a larger phenomenological study which explored the experiences of Novice Nurse Educators (NNEs) transitioning from clinical practice to academia in Ghana. Given the lack of evidence on the phenomenon within the Ghanaian context, this specific paper aimed to explore the reasons for NNEs' transition from practice to academia in Ghana. Understanding these reasons is

imperative, as it will potentially guide the development of a comprehensive tutor recruitment policy for the employment and retention of competent nurse educators in Ghana.

## Materials and methods

### Study design and setting

The study was conducted using a descriptive phenomenology design to explore the reasons for NNEs' transition from practice to academia. The design allowed the researchers to explore the phenomenon as experienced by the participants without interpretation [19]. The Consolidated Criteria for Reporting Qualitative research (COREQ) checklist S1 File guided the write-up [20]. The study was conducted in the Upper East Region of Ghana. The Upper East Region is one of Ghana's sixteen administrative regions located in the north-eastern corner of the country with Bolgatanga as its capital. The region is largely (79%) rural and has five state-owned HTIs situated in four districts and municipalities. All the HTIs in the region are state-owned. Participants were recruited from three selected HTIs that offer a wide range of nursing and midwifery programmes with diverse teaching staff. The three HTIs were selected out of five based on their key characteristics in terms of the nursing programmes they offer (certificate, post-certificate, and diploma) and their locations in the rural, peri-urban, and urban areas of the region, which offers a comparison of the topic in these settings.

### Population, sampling technique, and sample size

NNEs in the three selected HTIs constituted the study population. The inclusion criteria were nurse educators who had had a minimum of three years of past clinical practice experience before becoming health tutors, had been teaching for less than three years in the selected institutions, and were willing to participate in the study. Participants were selected using the purposive sampling technique which permitted the researchers to select participants based on the researchers' judgement about which participants met the inclusion criteria [21]. The researchers visited the selected HTIs to identify those who met the inclusion criteria. The researchers had no prior relationships with the participants and thus, did not influence the selection of study participants. There was no selection bias as participants were selected strictly based on the inclusion criteria. The study was explained in detail and potential participants were given information sheets containing further explanation about the study. Only those who willingly agreed to participate in the study and gave written informed consent were added to the study. No participant dropped out of the study. Thus, twelve NNEs were recruited for the study which was determined by data saturation as data redundancy occurred with the 12th participant [22].

### Data collection tool and procedure

The data collection method was individual face-to-face in-depth interviews through a semi-structured interview guide S2 File developed by the researchers. The interview guide covered areas including participants' profile and reasons for transitioning from practice to academia. The interview guide had questions such as; please tell me the reason why you left clinical nursing practice, what were your expectations before moving to nursing academia, what factors attracted you to nursing academia, what informed your decision to become a nurse educator? The questions and probes were developed based on a review of relevant literature and the study objectives. Two weeks before the actual data collection, the interview guide was piloted with 3 NNEs in one of the selected HTIs to sharpen interviewing skills, detect flaws in the guide, and make amendments to guarantee reliability and precision [23]. None of the

participants in the pilot study was recruited into the actual study and the results of the pilot study were not added to the actual study results. All the interviews were conducted by the first author, a male qualitative nursing researcher with rich experiences in nursing practice and education. Four interviews were conducted in each health training institution between June and August 2020. Two interviews were conducted each week in the offices of participants usually after lectures (determined by participants) and lasted between sixty and ninety minutes. Only the first author and a participant were in the office during each interview. All interviews were audio-recorded using a digital voice recorder with permission and important gestures and non-verbal observations were captured in a field notes book. During interview sessions, questions were asked about the reasons for transitioning from practice to academia. Open-ended questions were asked with follow-up probes to obtain an in-depth understanding of participants' narrations [24]. No repeat interviews were conducted.

## Data analysis and management

Data were analysed manually and concurrently with data collection which was guided by Colaizzi's approach to descriptive phenomenology data analysis [25]. The first author transcribed each audio-recorded interview verbatim. The first and second authors bracketed their minds and applied Colaizzi's approach as detailed below.

*Step 1*: The authors familiarised themselves with the data by reading through each transcript several times to obtain a general sense of the data.

*Step 2*: For each transcript, all statements and phrases about NNEs' reasons for transitioning from practice to academia were identified.

*Step 3*: Meanings were formulated for the identified statements and phrases about the reasons for transitioning from practice to academia.

*Step 4*: The formulated meanings were then categorised into clusters of themes that are common across all the transcripts. Field notes were also used to corroborate the emergent themes.

*Step 5*: A rich and exhaustive description of NNEs' reasons for transitioning from practice to academia was written to incorporate all the themes.

*Step 6*: The exhaustive description was then condensed down to short statements that represented the reasons for transitioning from practice to academia.

*Step 7*: The extracted statements were then returned to the participants for validation (member checking). However, no relevant information was obtained at this step.

Data were managed manually as soft copies of the transcripts were saved in a secured folder in the password-protected computer of the first author. Copies of the transcripts were also saved in a secured pen drive to prevent data loss and ensure easy retrieval of data. Also, codes were assigned to each transcript to ensure the anonymity of participants.

## Trustworthiness

Lincoln and Guba's criteria of credibility, transferability, dependability, and confirmability were applied to ensure trustworthiness [26]. By using the purposive sampling method, only participants who had relevant experience on the subject of study were enrolled in the study. Member checking was done to ensure credibility where the findings were returned to the participants for validation before drawing conclusions on the data [27]. Thick description was

also done to ensure transferability where the study setting, methodological processes, and sample characteristics were vividly detailed [28]. Peer debriefing was also done where the researchers had sessions with experienced colleagues to review all aspects of the study. By mind bracketing and maintaining a reflexive journal to prevent bias and enhance data objectivity, confirmability was ensured [29].

### Ethical considerations

The Committee on Human Research, Publication, and Ethics (CHRPE) at the Kwame Nkrumah University of Science and Technology (KNUST) gave ethical approval for the study with reference number CHRPE/AP/195/20. All three HTIs also gave institutional approvals before data collection started. All participants were given a detailed explanation of the study including the assurance of confidentiality and the right to participate and withdraw with no consequences or penalties. All participants gave written informed consent by signing the informed consent forms before data collection commenced. By signing the consent form, they consented to voluntarily participate in the study, be interviewed, and audio recorded. Participants' personal identifying information was omitted from the transcripts and replaced with codes (NNE1...NNE12) to ensure the anonymity of participants.

## Results

### Demographic characteristics of participants

Twelve NNEs participated in the study including two females and ten males. Their ages ranged between 32 and 39 years with an average age of 34.4 years. All the participants ($n$ = 12) were married. Regarding the highest degree attained, nine participants had a bachelor's degree in nursing, two had a bachelor's degree in public health, and one had a bachelor's degree in midwifery. The participants had between 6 and 10 years of previous clinical practice experience averaging 7.25 years. Participants also had either 1 or 2 years (average 1.75 years) of teaching experience. The detailed demographic characteristics of the participants are shown in C:\Users \lyrsc\Downloads\S1_Table.docxS1 Table.

### Main findings

Four themes, each eliciting NNEs' reasons for transitioning from practice to academia emerged from the data and include (1) dissatisfied with clinical nursing, (2) quest for flexible work role, (3) quest for work autonomy, and (4) previous clinical teaching. Table 1 presents the themes generated from the data.

**Dissatisfied with clinical nursing.** Dissatisfaction in the clinical setting was a common experience mentioned in the study as some participants disclosed that they were not satisfied with their job in the clinical area. Participants' dissatisfaction mostly emanated from frustration, denial of opportunity to further their education, poor nursing leadership, and long hours of work. This dissatisfaction eventually became the reason why participants transitioned from

**Table 1. Themes generated from the data.**

| Themes | |
|---|---|
| Theme 1 | Dissatisfied with clinical nursing |
| Theme 2 | Quest for flexible work role |
| Theme 3 | Quest for work autonomy |
| Theme 4 | Previous clinical teaching |

clinical practice to academia. A participant narrated how frustration in his previous nurse-clinician role resulted in him being dissatisfied in the clinical setting.

> *"I felt the clinical setting was frustrating because I was working so hard and yet, no one saw the work that I was doing. Whatever knowledge I had, sometimes the doctors and the other staff did not really appreciate it, when I give some other opinions, they feel like I was too knowing. So I felt coming this way will be appropriate because I am doing basically what I am supposed to do without frustration" (NNE1).*

Other participants recounted how they were dissatisfied at the clinical setting after they were denied the opportunity to further their education. This is reflected in the following quotations.

> *"I was not satisfied at the maternity ward because I was denied the opportunity to further my education because the matron did not want me to go to school. At the time, I had worked for five years of continuous service at the ward yet, I was denied study leave with pay whiles my juniors who barely worked for three years were all allowed to go to school" (NNE4).*

> *"Honestly, the dissatisfaction and frustration in the clinical setting influenced me to move into nursing academia. After several years of working in the ward, I still didn't get study leave so I moved to this place [academia] so that I can get the chance to further my education" (NNE10).*

Other participants also narrated their ordeal about the dissatisfaction at the clinical setting and associated their dissatisfaction with poor leadership in the clinical setting.

> *"Looking back, I was not happy in that hospital because my ward in charge and even the matron of the hospital in which I worked were not good leaders. My ward in charge was a dictator and never allowed we the subordinates to verbalise our concerns. Likewise, the hospital matron was also authoritative and hardly listened to opinions from us. Their leadership styles were bad and I couldn't continue working in such an environment" (NNE2).*

> *"One day I went to work late, about 30 minutes late because I had a tire burst on my way to work. Surprisingly, I was insulted by my ward in charge and denied the opportunity to explain myself. She was just a dictator" (NNE7).*

Additionally, a participant narrated how his long work hours made him dissatisfied with his role in the clinical setting.

> *"When I was working at the paediatrics ward, I could work for nine to ten hours before I closed and this really made me exhausted, and I gradually lost interest in the clinical setting. So I moved to this place [college] to relax small" (NNE12).*

**Quest for flexible work role.**   Participants generally mentioned that there was a lack of role flexibility in the clinical setting which affected their lifestyle. They felt the nurse educator role was more flexible and translated into a flexible lifestyle. Therefore, the flexibility of the nurse educator role has been the reason why some participants chose to transition from the bedside to the classroom. A participant noted that running shifts at the ward not flexible and that informed his decision to transition to academia.

*"Working in the ward was not flexible at all because like other nurses, I often went to work either for the morning shift, afternoon shift, or night shift and I was expected to follow the roster meticulously. Therefore, I needed a more flexible work role, and that informed my decision to move to this place [college]" (NNE6).*

Both female participants of the study also wanted a more flexible work role because they were married and wanted to be free of night duties and to spend time with their spouses and family at home at night.

*"Me I am married so I wanted a job that would not demand that I work at night because I wanted to always spend my nights with my husband and children at home. And since night shift was a constant thing there [at the clinical setting], I decided to move to this place [college]" (NNE11).*

*"When I was at the maternity ward I used to go for night duties every month and that created a lot of confusion between me and my husband. So based on that, I decided to go into teaching so that I will be free from night duties and always spend the night with my husband and family at home" (NNE5).*

Other participants also wanted work role flexibility because of weekends. They wanted to be free of work on Sundays due to church service and to spend the weekends with their families.

*"I am a strong Christian so I don't joke with my Sundays at all because I a leader in the church and so I have to always be in church on Sundays. Since it was not possible to have all the Sundays to myself while at the ward, I decided to run to the teaching side so that I will always be able to go to church on Sundays" (NNE4).*

*"I wanted a flexible work which will not call me to work on weekends especially on Saturdays because, I really enjoy spending my weekends with friends and family, so I moved to the school here" (NNE8).*

**Quest for work autonomy.** There is a direct link between work autonomy and job satisfaction and intention to stay. In this study, however, participants recounted working in the clinical setting without work autonomy but perceived the nurse educator role as highly autonomous. Therefore, the quest for autonomy was a decisive reason that motivated participants to transition into nursing academia. Some participants narrated how medical doctors and senior nurses constantly supervised their work in the clinical setting.

*"After about 6 years of midwifery practice at the maternity unit, I got to realise that nurses and midwives were always working under medical doctors and in most instances, we were virtually servants to medical doctors. I felt l lacked the independence to perform my duties as a midwifery officer because I was not allowed to do anything in the maternity ward without the supervision of a gynaecologist. Therefore, I decided to move into teaching since there is autonomy here [college]" (NNE5).*

*"The senior nurses were always on us, [clinical nurses] monitoring and supervising whatever we [clinical nurses] did in the ward as if we [clinical nurses] were not competent enough. I actually felt inferior at the clinical setting and I obviously couldn't continue in that tangent" (NNE6).*

A participant also narrated how he was not allowed to perform some nursing procedures like catheterisation and surgical wound dressing without the supervision of doctors and senior nurses.

*"I was not allowed to carry out some nursing procedures independently. I remember some time ago in the surgical ward, I was not even allowed to pass a urethral catheter without supervision from the doctor [laughing at the recollection] and we [clinical nurses] were not allowed to even perform bladder irrigation, not to talk of minor suturing and prescription" (NNE10).*

*"Our ward in-charge was always on our neck [laughing at the recollection] supervising and giving unnecessary directives on the ward. She was always all over the ward checking everything we did, trying to find faults. Me too I hate to be witch-hunted [laughing] so I decided to move here [school] to be free" (NNE1).*

**Previous clinical teaching.** Another theme relevant to the reasons for transitioning from practice to academia was previous clinical teaching. For all participants, they were already doing some form of teaching in the clinical setting when they were nurse clinicians and that motivated them to go into academia. Two participants were previously preceptors for students at the ward and taught nursing students during their vacation practicum and that formed the basis for them to decide to go into academia.

*"In those days at the medical ward, I was the focal person for nursing students, so whenever students came for vacation practicum, I taught them all the nursing procedures they were mandated to learn during the practicum period. That was where I developed an interest in teaching, and I felt it would be better for me to move into the academic side so that I can teach full-time" (NNE9).*

*"I worked in the medical and surgical wards for seven years, and when I was in the surgical ward, I was a preceptor for nursing students. I used to teach them a lot of things like bed making, bed bath, serving bedpans, and the rest, and whenever I finished a teaching session, some of them [students] told me that I taught even better than their tutors in the school. So based on this, I decided to just move into teaching since I was good at it" (NNE3).*

Two public health nurses also cited their previous client teachings on topics like family planning and exclusive breastfeeding motivated them to transition into full-time teaching.

*"I was a public health nurse by then, and you know I was doing a lot of patient teaching on FP [Family Planning]. I found a lot of joy and happiness during my teaching sessions with my clients, so when the opportunity came for me to go into teaching on a full-time basis, I didn't hesitate" (NNE11).*

*"Back then I used to give a lot of education to our clients at the public health unit of the hospital, the clients were always many and the in-charge used to select me to give education on exclusive breastfeeding [laughing at the recollection]. So small, small, small, I just decided to move here to teach students" (NNE4).*

For some participants, educating patients on their disease condition and medication regimen was what prompted them to transition from the bedside to the classroom.

*"In the ward, I used to educate patients on their disease conditions and medications, and the clients understood me very well so I decided to move to this place so that I can do the teaching officially" (NNE2).*

In summary, participants in this study talked about their reasons for transitioning from practice to academia. It appears from the results of our study that NNEs decided to transition from practice to academia mostly because of their previous negative experiences in the clinical setting. They transitioned because they were dissatisfied with clinical nursing, wanted more flexible work, wanted to work autonomously, and previously taught in the clinical setting.

## Discussion

This study identified that NNEs in the Upper East Region transitioned from practice to academia because they were dissatisfied with their previous nurse clinician role in the hospital setting. For nurses to perform optimally and give their best in the care of patients, they need to be satisfied with the work they do. Yet, participants reported that frustration at the clinical setting, denial of study leave, poor nursing leadership, and long work hours at the clinical area contributed to their dissatisfaction and informed their decision to transition to academia. Our findings concur with a study in the United States of America (USA) that found that novice educators decided to transition from practice to academia because of dissatisfaction and frustration with the clinical systems, 12-hour shifts, tired of the commute to work, and feelings of burn out [30]. However, contrary to our findings, several other studies in the literature have found that NNEs transitioned from clinical practice to academia because of good reasons like having an innate desire and love for teaching [11] and not merely being dissatisfied with their previous practice role. When nurses transition from the clinical setting to academia for the right reasons such as having the desire to share nursing knowledge, it leads to job satisfaction and retention [9]. Therefore, for NNEs in this study to transition from practice to academia only because they were dissatisfied with their role in the clinical setting has some implications for recruiting and retaining committed educators in our HTIs. The HTIs secretariat in Ghana should reconsider their method of recruiting health tutors and factor in measures to ensure that nurses are recruited from the clinical setting to the academic setting for the right reasons.

The flexibility of the nurse educator role, as well as the quest for a flexible work role, motivated some participants to transition from practice to academia. It was noted that their work role in the clinical setting was not flexible enough due to the shift system of work, night duty schedules, and working during weekends. This finding echoes the findings of other researchers who reported that nurse educators transitioned from practice to academia in search of a more flexible work schedule and flexibility in scheduling [12,15]. Our findings are in agreement with a study conducted in the United States of America (USA) which found that NNEs transitioned from practice to academia because they wanted work role flexibility to balance work role and family lifestyle [31]. Our findings also concur with a study that revealed that the want for a more flexible work schedule and vacation time were the reasons for NNEs transition to academia in Canada [16]. Some distinct findings in our study are that all nurse educators were married and desired to be exempted from night duties to spend time with their spouses and children at night at home, others were ardent Christians and wanted to be free of work on Sundays so they could devote much time for church, and some wanted to be free on weekends to spend quality time with friends and family. Hence, they decided to transition to academia because of the perceived work role flexibility associated with the nurse educator role. However, this finding has an implication for the recruitment and retention of committed nurse educators because, despite the illusion of shorter days, flexibility, and freedom to enjoy weekends

and holidays, academia actually requires a significant time commitment [7] and NNEs may enter into a state of transition shock upon realization of this reality.

Participants in this current study decided to transition from practice to academia because they wanted work autonomy. NNEs in this study felt they lacked work autonomy during their professional nursing practice at the clinical setting and perceived the nursing academia as a role with abundant autonomy, and hence, their reason for transitioning from practice to academia. This finding is consistent with a study in Australia and the United Kingdom (UK) where the move into nursing academia revealed a working environment that had more autonomy and academic freedom for Australian and UK nurse educators [32]. In the United States of America (USA), work autonomy and independence have also been associated with the nurse educator role [12]. Autonomy to work plays an essential role in job satisfaction and retention among nurses [33]. Unsurprisingly, therefore, the lack of autonomy in clinical nursing practice, as well as the quest for autonomy in the nursing faculty role, was central to participants' decision to transition from practice to academia. A significant finding in the current study is that participants were constantly supervised by medical doctors and senior nurses in the clinical setting when performing nursing procedures like catheterisation and surgical wound dressing. This constant supervision from medical doctors and senior nurses eventually became pivotal to participants' departure from the clinical setting. This finding also highlights the fact that NNEs transitioned from practice to academia because of their previous negative experience in the clinical setting and not necessarily because of positive reasons. This implies the need for principals of HTIs in Ghana, as well as the Ministry of Health (MOH) through the HTIs secretariat to evaluate the health tutor recruitment process to ensure that clinical nurses with positive reasons are recruited into the various HTIs as tutors.

For NNEs in this study, having previous clinical teaching experience was a big motivation for their transition from practice to academia. This finding is consistent with a study in Canada where Canadian nurse educators previously taught as part of their nursing practice, and these teaching experiences were the most satisfying parts of their practice as nurses and were a consideration in deciding to work as nurse academics [16]. A significant finding in the current study is that some NNEs reported that they were previously preceptors for students at the ward and taught nursing students during their vacation practicum, others gave education on family planning and exclusive breastfeeding at the public health units of their hospitals, and some educated their patients on disease condition and medication regimen while at the clinical setting. These previous clinical teaching experiences ultimately motivated participants to transition into full-time teaching at the college level. However, this finding comes as no surprise because participants in the current study had immense previous clinical practice experience, thus, between 6 and 10 years (average 7.25 years).

## Limitations of the study

The authors acknowledge some limitations in the study. First, only one midwifery tutor was included in the study. Thus, the inadequate number of midwifery tutors in the study was a limitation since the inclusion of more midwifery tutors could generate new findings that could have altered the study findings. Also, only two female nurse educators were included in the study, which could limit the transferability of the study findings. Notwithstanding, this study provides useful information on NNEs' reasons for transitioning from practice to academia in Ghana.

## Conclusion

The study revealed that NNEs in the Upper East Region of Ghana had various reasons for transitioning from clinical practice to nursing academia including being dissatisfied with clinical

practice, wanting a more flexible work role, wanting to work autonomously, and have previously taught at the clinical setting. These reasons mostly reflect NNEs' negative experiences in the clinical setting which have serious implications on the recruitment and retention of the right teaching staff to train the next generation of nurses. The findings of the study suggest the need to revise and implement a tutor recruitment policy that takes into account, what attracts clinical nurses to academic citizenship. Hence, Ghana's MOH through the HTIs secretariat should use the findings of this study as a foundation to facilitate the development of creative policies and strategies to identify, recruit, and retain qualified nurse educators.

## Supporting information

**S1 Table. Demographic characteristics of participants.**
(DOCX)

**S1 File. COREQ checklist.**
(PDF)

**S2 File. Semi-structured interview guide.**
(DOCX)

## Acknowledgments

The authors are grateful to the nurse educators for participating in the study. We are also grateful to the principals of the three HTIs for giving administrative approvals.

## Author Contributions

**Conceptualization:** Timothy Tienbia Laari, Felix Apiribu.

**Data curation:** Timothy Tienbia Laari.

**Formal analysis:** Timothy Tienbia Laari, Felix Apiribu.

**Investigation:** Timothy Tienbia Laari, Felix Apiribu.

**Methodology:** Timothy Tienbia Laari, Felix Apiribu, Philemon Adoliwine Amooba.

**Project administration:** Timothy Tienbia Laari, Felix Apiribu.

**Resources:** Timothy Tienbia Laari.

**Supervision:** Felix Apiribu.

**Validation:** Timothy Tienbia Laari, Felix Apiribu.

**Writing – original draft:** Timothy Tienbia Laari, Felix Apiribu.

**Writing – review & editing:** Timothy Tienbia Laari, Felix Apiribu, Philemon Adoliwine Amooba, Adwoa Bemah Boamah Mensah, Timothy Gazari, Joseph Kuufaakang Kuunibe, Gideon Awenabisa Atanuriba, Moses Haruna Akor.

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
