## [Decision Letter · Decision Letter 0]

17 Aug 2021

PONE-D-21-16603

Exploring the reasons for novice nurse educators’ transition from practice to academia in Ghana

PLOS ONE

Dear Dr. Laari,

Thank you for submitting your manuscript to PLOS ONE. After careful consideration, we feel that it has merit but does not fully meet PLOS ONE’s publication criteria as it currently stands. Therefore, we invite you to submit a revised version of the manuscript that addresses the points raised during the review process.

We look forward to receiving your revised manuscript.

Kind regards,

Khatijah Lim Abdullah, DClinP, MSc., BSc

Academic Editor

PLOS ONE

Journal Requirements:

Reviewers' comments:

**Comments to the Author**

Reviewer #1: Dear Author(s),

Thank you for submitting this article. I believe that you have put a huge effort to come out with this paper. Congratulations. Here are my minor comments that you may want to consider:

1. While you managed to present the study background and discussion in an acceptable manner, the majority of the cited studies came from developed countries (e.g US, Canada, UK, Australia). Adding some information on experience of novice nurse educator in low middle-income countries would be helpful to set up background of the study/ improve the the discussion section.

2. The abbreviation of “HTI” was inconsistently used with some abbreviation and some full spelling.

Thank you.

Reviewer #2: Review comments to author:

First of all, congratulations on writing a paper on a subject that may not only apply to the nursing profession, but also to others and the perspective of Ghana is directly related to policy implementation.

However, I do have some burning questions needing answers to and this may require some revision from the authors:

1. Has the manuscript underwent the qualitative research checklist? The COREQ 32-item checklist should be adhered and submitted along as part of the submission as they will become part of the following questions inquiries.

2. Introduction: please describe the nursing profession in Ghana to give an idea of the characteristics of the profession, composition and tie-ins to the issues highlighted in the manuscript.

3. Line 123-125: It was stated there were 5 state-owned training centres located in the 4 regions. Are there non-state owned training centres and if they are, why are they not included in the study? If there are only state-owned training centres, why are only 3 selected? Why not all 5? Please justify.

4. Line 131-133: Why are researchers selecting “the most informative” participants? How does the study handle selection bias for this study? Does the researcher have prior relationships with the participants? How do they identify participants who are “most informative”?

5. Line 141-142: What are the domains to be explored of the developed IDI guides and what probes are used to further explore those issues? How were they developed? Kindly explain in detail.

6. Line 143-145: Was the pilot done in the same 3 institutions and were the same pilot participants recruited in the actual data collection? When was the pilot conducted and IDI guide revised?

7. Line 191-192: When was consent taken and what is the coverage of this consent? Is it just for the interview or did it cover anything else? Please detail further.

8. Line 287-292 & 325-329: The two highlighted quotes coming from the same NNE5 sounds contradicting. In line 287-292, NNE5 claims not being allowed to do anything without supervision but line 327-329, NNE5 claims to enjoy freedom in teaching clients about antenatal care. So there is a degree of autonomy that is actually given but claim without by the same participant. Kindly revise.

9. Line 362: “wanting to make a difference in nursing (13), and to educate the next generation of nurses (11)” – there are no verbatims in any of the 4 domains in the results that supported these two reasons. Please revise or add the supporting verbatims and statements in the results section.

10. Conclusion – With the concluded results, how would the information is to be used by the relevant stakeholders and what action should be recommended to tackle the issue?

---

## [Author Response · Author response to Decision Letter 0]

23 Aug 2021

Dear reviewers, we are grateful to you for making time to review our manuscript. We believe that your expert reviews and comments have enhanced the quality of our paper. Responses have been provided below each comment. 

Thank you. 

Response to Reviewer #1

1. While you managed to present the study background and discussion in an acceptable manner, the majority of the cited studies came from developed countries (e.g US, Canada, UK, Australia). Adding some information on experience of novice nurse educator in low middle-income countries would be helpful to set up background of the study/ improve the discussion section.

Response

Some information on experiences of novice nurse educators in Low and Middle-Income Countries (LMICs) have been added to the background.

2. The abbreviation of “HTI” was inconsistently used with some abbreviation and some full spelling.

Response

The abbreviation “HTIs” has now been consistently used throughout the manuscript except in the abstract and title.

Response to Reviewer #2

1. Has the manuscript underwent the qualitative research checklist? The COREQ 32-item checklist should be adhered and submitted along as part of the submission as they will become part of the following questions inquiries.

Response

The Consolidated Criteria for Reporting Qualitative research (COREQ) 32-item checklist has now been adhered to. It has also been added to the submission as a supporting information (S1 File).

2. Introduction: please describe the nursing profession in Ghana to give an idea of the characteristics of the profession, composition and tie-ins to the issues highlighted in the manuscript.

Response

A detailed description of the nursing profession and education in Ghana has been provided in the introduction.

3. Line 123-125: It was stated there were 5 state-owned training centres located in the 4 regions. Are there non-state owned training centres and if they are, why are they not included in the study? If there are only state-owned training centres, why are only 3 selected? Why not all 5? Please justify.

Response

The three HTIs were selected out of five based on their key characteristics in terms of the nursing programmes they offer (certificate, post-certificate, and diploma) and their locations in the rural, peri-urban, and urban areas of the region, which offers a comparison of tutor experiences in these settings.

4. Line 131-133: Why are researchers selecting “the most informative” participants? How does the study handle selection bias for this study? Does the researcher have prior relationships with the participants? How do they identify participants who are “most informative”?

Response

“Most informative participants” in our view are participants who met the inclusion criteria. However, “most informative participants” appears not to best fit in the paper. Therefore, it has been replaced with “participants who met the inclusion criteria”. 

The researchers had no prior relationships with the participants and thus, had no influence on the selection of study participants. 

There was no selection bias as participants were selected strictly based on the inclusion criteria.

5. Line 141-142: What are the domains to be explored of the developed IDI guides and what probes are used to further explore those issues? How were they developed? Kindly explain in detail.

Response

The interview guide covered areas including participants’ profile and reasons for transitioning from practice to academia. The interview guide had questions such as; please tell me the reason why you left clinical nursing practice, what were your expectations before moving to nursing academia, what factors attracted you to nursing academia, what informed your decision to become a nurse educator? 

The questions and probes were developed based on a review of relevant literature and the study objectives

6. Line 143-145: Was the pilot done in the same 3 institutions and were the same pilot participants recruited in the actual data collection? When was the pilot conducted and IDI guide revised?

Response

Piloting was done in one of the selected institutions and the interview guide revised two weeks before the actual data collection. None of the participants in the pilot study was recruited into the actual study and the results of the pilot study were not added to the actual study results.

7. Line 191-192: When was consent taken and what is the coverage of this consent? Is it just for the interview or did it cover anything else? Please detail further.

Response

Consent was taken before data collection commenced. By signing the consent form, they consented to voluntarily participate in the study, be interviewed, and audio recorded.

8. Line 287-292 & 325-329: The two highlighted quotes coming from the same NNE5 sounds contradicting. In line 287-292, NNE5 claims not being allowed to do anything without supervision but line 327-329, NNE5 claims to enjoy freedom in teaching clients about antenatal care. So there is a degree of autonomy that is actually given but claim without by the same participant. Kindly revise.

Response

Comparing the two statements made by the same participant (NNE5), it appears that she narrated her experience from two separate units in the hospital. The former is an account by a midwife (NNE5) who lacked work autonomy in the “maternity ward”. The latter is also an account by the same midwife (NNE5) who taught her clients “during antenatal care (ANC) and postnatal care (PNC)”. However, we agree that there is an iota of contradiction in those statements. Therefore, the latter has been removed from the manuscript.

9. Line 362: “wanting to make a difference in nursing (13), and to educate the next generation of nurses (11)” – there are no verbatims in any of the 4 domains in the results that supported these two reasons. Please revise or add the supporting verbatims and statements in the results section.

Response

After a careful consideration, this portion of the manuscript has been revised and “wanting to make a difference in nursing (13), and to educate the next generation of nurses (11)” has been removed from the manuscript.

10. Conclusion – With the concluded results, how would the information is to be used by the relevant stakeholders and what action should be recommended to tackle the issue?

Response

Ghana’s Ministry of Health (MOH) through the HTIs secretariat should use the findings of this study as a foundation to facilitate the development of creative policies and strategies to identify, recruit, and retain qualified nurse educators.

Response to Editor comments

PLOS ONE's style requirements, including those for file naming have been looked at and complied with. Our reference list has also been reviewed to ensure that it is complete and correct.

CHANGES TO THE REFERENCE LIST

Due to the additional information provided in the introduction, five (5) references have been added to the reference list and appropriately cited in the text.

---

## [Decision Letter · Decision Letter 1]

4 Oct 2021

Exploring the reasons for novice nurse educators’ transition from practice to academia in Ghana

PONE-D-21-16603R1

Dear Dr. Laari,

We’re pleased to inform you that your manuscript has been judged scientifically suitable for publication and will be formally accepted for publication once it meets all outstanding technical requirements.

Kind regards,

Khatijah Lim Abdullah, DClinP, MSc., BSc

Academic Editor

PLOS ONE

---

## [Editor Report · Acceptance letter]

7 Oct 2021

PONE-D-21-16603R1 

Exploring the reasons for novice nurse educators’ transition from practice to academia in Ghana 

Dear Dr. Laari:

I'm pleased to inform you that your manuscript has been deemed suitable for publication in PLOS ONE. Congratulations! Your manuscript is now with our production department. 

Kind regards, 

on behalf of

Dr. Khatijah Lim Abdullah 

Academic Editor

PLOS ONE